# Prevalence of ESβL, AmpC and Colistin-Resistant *E. coli* in Meat: A Comparison between Pork and Wild Boar

**DOI:** 10.3390/microorganisms9020214

**Published:** 2021-01-21

**Authors:** Martina Rega, Ilaria Carmosino, Paolo Bonilauri, Viviana Frascolla, Alice Vismarra, Cristina Bacci

**Affiliations:** 1Food Hygiene and Inspection Unit, Department of Veterinary Science, University of Parma, Strada del Taglio, 10, 43126 Parma, Italy; martina.rega@unipr.it (M.R.); ilaria.carmosino@gmail.com (I.C.); frviviana@hotmail.it (V.F.); cristina.bacci@unipr.it (C.B.); 2Istituto Zooprofilattico Sperimentale della Lombardia ed Emilia Romagna, 42124 Reggio Emilia, Italy; paolo.bonilauri@izsler.it; 3Parasitology Unit, Department of the Veterinary Science, University of Parma, Strada del Taglio, 10, 43126 Parma, Italy

**Keywords:** ESβL, AmpC, *E. coli*, pork, wild boar, meat, colistin, *bla*_CTX-M1_, *mcr-1*

## Abstract

A global increase in *Escherichia coli* (*E. coli*) resistant to cephalosporins (extended-spectrum β-lactamases (ESβLs) and AmpC β-lactamases) has been recorded in the last 20 years. Similarly, several studies have reported the spread of colistin resistance in Enterobacteriaceae isolated from food and the environment. The aim of the present study was to evaluate the prevalence of ESβL, AmpC and colistin-resistant *E. coli* isolated from pork and wild boar meat products in the Emilia Romagna region (North Italy). The isolates were analysed phenotypically (considering both resistant and intermediate profiles) and genotypically. The prevalence of genotypically confirmed ESβL and AmpC *E. coli* was higher in pork meat products (ESβL = 11.1% vs. AmpC = 0.3%) compared to wild boar meat (ESβL = 6.5% vs. AmpC = 0%). Intermediate profiles for cefotaxime (CTX) and ceftazidime (CAZ) were genotypically confirmed as ESβL in pork meat isolates but not for wild boar. Four *E. coli* from wild boar meat were resistant to colistin but did not harbour the *mcr-1* gene. *E. coli* isolated from wild boar meat seem to show aspecific antimicrobial resistance mechanisms for cephalosporins and colistin. The prevalence of resistant isolates found in wild boar is less alarming than in pork from farmed domestic pigs. However, the potential risk to consumers of these meat products will require further investigations.

## 1. Introduction

Antimicrobials are necessary agents to fight diseases in humans, animals, plants and crops. Despite this, their use is complicated by the development of antimicrobial resistance (AMR) [1]. One of the major causes of this natural process is the overuse of antibiotics and their inappropriate administration (wrong category of antibiotic, inadequate dose, and reduced duration of therapy) [2], and this results in a high quantity of failed treatments against pathogens and in increasing mortality [3]. AMR has been detected particularly among commensal gut bacteria with patterns that vary across species, locations and times [4]. For that reason, since 2008, the World Organization for Animal Health (Oie) has published guidelines with the aim to encourage prudent use of antimicrobials [5]. 

It is estimated that resistance to second and third generations of cephalosporins will double by 2030 [3]. Human infections caused by cephalosporin-resistant *Escherichia coli* (*E. coli*), for example, are increasing, and a significant risk factor is represented by the use of these drugs in farming [3]. *E. coli* become resistant to cephalosporins due to their ability to produce β-lactamases, enzymes that hydrolyse the β-lactam ring and deactivate the molecule [6]. 

*E. coli* that produce extended-spectrum β-lactamases (ESβLs) or AmpC β-lactamases have been isolated from different food-producing animals, including calves, cattle, swine, broilers, horses and rabbits, in several countries in the EU, suggesting a possible role of animals and food as important reservoirs [7]. 

Another emerging risk is represented by bacteria resistant to colistin, also known as polymyxin E [8,9]. Colistin is still active against multidrug-resistant (MDR) Gram-negative pathogens and most of the members of Enterobacteriaceae family, including *E. coli* [10]. For this reason, it is considered a last line of treatment for human infections [9] that do not respond to other antibiotics. The mechanism of resistance is based on the transfer of phosphoethanolamine (PEtN) transferase to lipid A, resulting in reduced anionic charges on the bacterial membrane and a reduction of the ability of colistin to interact with the microorganism [11]. The mechanism seems to be related to the expression of *mcr* genes [10], generally located on plasmids; *mcr-1* to *mcr-9* gene variants have been identified [12]. The *mcr-1* gene is the most widespread of these gene variants in Europe and was also identified in *E. coli* and other bacteria (such as *Salmonella enterica*) isolated from pigs and chickens. The universal distribution of the *mcr-1* gene suggests a potential food chain dissemination pathway due to its presence in food, animals and the human microbiome [13].

The spread of AMR is considered an important public health risk and is related both to pathogenic and commensal microorganisms. For example, *E. coli*, an intestinal commensal microorganism, can sometimes cause community-onset infection even in healthy people [14] or can be a reservoir of resistance genes for pathogenic bacteria [15]. In the last few decades, studies have reported an increase in the prevalence of multidrug-resistant coliform bacteria, not only in hospital settings, but also in swine herds (*Sus scrofa domesticus*) [14], wild animals (particularly the wild boar, *Sus scrofa*) and in their meat products [16].

Antimicrobials used in food-producing animals (such as swine) represent more than 70% of the antimicrobials produced worldwide. They are used for treating, controlling and preventing diseases, but they have also been used for their effect in improving growth performance [17].

Antimicrobials used to treat both human and domesticated animals can be dispersed in the environment (in sewage effluent pumped into rivers or in the spread of sewage sludge or animal sludge as fertiliser), and many can be also excreted in active form, placing selective pressure on environmental bacteria [18]. This is particularly true in heavily populated areas where there are many sources and amplifiers of AMR and could be associated with AMR commensal bacteria isolated from wild animals [19] such as wild boars [20], foxes [21] and birds [22]. Wild animals can carry AMR genes in their gut, and ineffective waste management and long-range animal movements may increase the risk of the spread of emerging AMR globally. 

AMR bacteria can also be transmitted through the food chain and particularly through the consumption of meat products [14,23]. Meat contamination is largely related to the process of slaughtering and/or hunting. Carcasses can be contaminated by manipulation, reflecting the hygienic level of the slaughterhouse and the animals’ health status and farming conditions [24]. It has been shown that good farming conditions and the respect of animal welfare are clearly correlated with a lower rate of animal disease and of bacterial microorganisms at the gut level (i.e., *Salmonella* and *Campylobacter*) [25]. 

In the literature, we can find a lot of data about the spread of ESβL and AmpC strains along the food chain; pigs are “under special surveillance” in this sense, but few studies are available for wild boars [16,20] and even fewer studies consider their meat. The diffusion of wild boars in flat land, close to human settlements and not only in rural areas, increased our interest in these animals and particularly in the fresh meat derived and, in some cases, slaughtered directly on the hunting set and not in accredited slaughterhouses. 

The aim of the present study was to define the prevalence of ESβL, AmpC and colistin-resistant *E. coli* in pork and wild boar meat products in order to define their potential transmission along the food chain and to evaluate any difference in resistance profile eventually present when considering their origin (farm vs. wildlife).

## 2. Materials and Methods

### 2.1. Sample Collection

Meat samples were collected by the Istituto Zooprofilattico Sperimentale della Lombardia e Emilia Romagna (IZSLER) located in Reggio Emilia (44°42′34″56 N, 10°37′13″80 E) between January 2018 and January 2020. The sample set was composed of a total of 1598 pork meat products (sausages, cotechino, salami, meatballs, meat skewers) from meat processing companies and 1052 fresh wild boar meat samples from slaughterhouses.

### 2.2. Escherichia coli Isolation and Counting

*Escherichia coli* were isolated from meat samples according to ISO 16649-2:2001. Briefly, 10 g of meat was homogenised 1:10 in Buffered Peptone Water (BPW) (Biolife Italiana, Milano, Italia); 1 mL was included in TBX agar medium (Biolife Italiana, Milano, Italia) and incubated at 44 ± 1 °C for 18–24 h. Typical blue-green colonies were selected and subjected to indole test. Positive colonies were confirmed as *E. coli* using API 20E miniaturised system (bioMérieux, Marcy-l’Étoile, France). Methods were accredited according to ISO 17025:2018 standard. The cutoff considered for further analysis was a number of *E. coli* colonies over 10 CFU/g.

*E. coli* isolates were then sent to the laboratory of Food Hygiene and Inspection of the Veterinary Science Department, University of Parma.

### 2.3. Antibiotic Susceptibility

All the isolated *E. coli* were tested for the ability to produce ESβL and AmpC, using the disk diffusion test following the protocol defined by EUCAST [26]. All the antibiotic disks were bought at Biolife Italiana, Milano, Italia. A screening test using two cephalosporins, cefotaxime 5 µg (CTX05) and ceftazidime 10 µg (CAZ10), was carried out. Isolates with a “resistance” (inhibition diameter CTX < 17 mm, CAZ < 19 mm) or “intermediate” profile (inhibition diameter 17 ≤ CTX < 20, 19 ≤ CAZ < 22) were phenotypically confirmed as ESβL and/or AmpC with the combination disk test (CDT) [26], using cefotaxime 30 µg (CTX30) or ceftazidime 30 µg (CAZ30) and cefotaxime (30 µg) or ceftazidime (30 µg) in combination with clavulanic acid (10 µg) (CTX + C and CAZ + C).

The confirmation of AmpC *E. coli* was performed through the application of cefotaxime 30 µg (CTX30) or ceftazidime 30 µg (CAZ30) and cefotaxime 30 µg + cloxacillin 10 µg (CTX + CX) or ceftazidime 30 µg + cloxacillin 10 µg (CAZ + CX).

Clavulanic acid and cloxacillin are two molecules that inhibit ESβL and AmpC activity, respectively.

The evaluation was done comparing the inhibition zone around the cephalosporin combined with the inhibiting molecule and the antibiotic alone.

Moreover, all the *E. coli* isolates were also tested for susceptibility to colistin using Sensititre plates (Thermofisher Scientific, Waltham, MA, USA) to define the minimal inhibitory concentration (MIC). Plates were prepared following the manufacturer’s instructions (Thermofisher Scientific, Waltham, MA, USA), and resistance to colistin was defined by an MIC > 2 µg/mL.

### 2.4. DNA Isolation and PCR for ESβL and AmpC Genes Identification

DNA from phenotypically confirmed ESβL and AmpC *E. coli* isolates was extracted using a commercial kit (Purelink genomic DNA purification kit, Invitrogen, Carlsbad, CA, USA) following the manufacturer’s instructions. In addition, *E. coli* isolates that resulted as being intermediate at the screening with cefotaxime and ceftazidime were also selected and analysed for the presence of ESβL and AmpC genes.

Briefly, an overnight broth culture was prepared starting from five pure colonies of *E. coli* grown on Tryptic Soy Agar (TSA) (Biolife Italiana, Milano, Italia), centrifuged at 15,000 rpm for 5 min, and the pellet was used for DNA purification using the reagents of the commercial kit mentioned above. Final elution was done in a volume of 50 µL with the elution buffer provided by the kit, and quantity and quality of the extracted DNA were determined with a spectrophotometer (Biospectrometer, Eppendorf, Hamburg, Germany).

A real-time PCR with Sybrgreen (SsoAdvanced SYBR Green Supermix Bio-Rad, Hercules, CA, USA) was applied in order to verify the presence of ESβL-associated genes: *bla*_CTX-M1_, *bla*_CTX-M2_, *bla*_TEM_ and *bla*_SHV_, as described by Roschansky et al. [27]. Preliminary tests to define the correct annealing temperature for each primer were done, and in each reaction positive (*K. pneumoniae* NCTC 13368 for *bla*_SHV_, *E. coli* NCTC 13351 for *bla*_TEM_ and *E. coli* NCTC 13353 for *bla*_CTX-M_) and negative controls were added. The presence of a nonspecific product was avoided through melting curve analysis. The amplification protocol was characterised by a denaturation step (95 °C for 3 min) and 39 repeated cycles (95 °C for 15 s; 50 °C for 15 s; 72 °C for 20 s). Fluorescence signals were collected in every cycle and each sample was tested four times.

The presence of AmpC genes was verified using the oligonucleotides and the multiplex PCR protocol described by Pérez-Pérez and Hanson [28], with the sole exception of the MgCl_2_ concentration (2 mM instead of 1.5 mM as suggested in the paper).

Isolates resistant to colistin were tested for the presence of the *mcr-1* gene following the end-point PCR protocol described by Liu et al. [29] with an optimised annealing temperature of 50 °C. Strain NCTC 13846 was used as a positive control.

### 2.5. Statistical Analysis

Statistical analyses were used to identify any significant differences between the variables considered. In particular, we compared the number of *E. coli* isolated from pork and wild boar meat, the number of ESβL genotypically confirmed between the two groups (pork vs. wild boar) and the same variables for all the ESβL and AmpC genes found. A chi-squared test was used to make these evaluations, and a *p*-value < 0.05 was considered as statistically significant.

Data collected had to respect the following relation to be considered statistically significant:n > 30; np > 5, n (1 − p) > 5(1)
where n is the number of animals and p is the proportion of *E. coli* strains with the characteristic under consideration in the study.

## 3. Results

### 3.1. Isolation of E. coli from Pork and Wild Boar Meat Products

Following the protocol described above and considering the cutoff of 10 CFU/g, a total of 314 pork meat samples (19.6%; CI 95% = 17.65–21.51) and 229 fresh wild boar meat samples (21.8%; CI 95% = 19.3–24.3) were contaminated by *E. coli*. In each contaminated sample, a single strain from the highest countable dilution was collected and forwarded for subsequent analysis. The comparison between the number of *E. coli* isolated from pork and that from wild boar meat showed a *p*-value of 1.15 (not statistically significant).

### 3.2. Antibiotic Resistance (ESβL and AmpC)

All the *E. coli* isolated were tested to define their susceptibility to third-generation cephalosporins, in particular to CTX and CAZ.

The phenotypic results showed that 7 *E. coli* pork meat isolates out of 314 (2.2%; CI 95% = 0.6–3.9) were ESβL, 3 (0.9%; CI 95% = 0–1.9) were AmpC and 5 (1.6%; CI 95% = 0.2–3.0) were both ESβL and AmpC. Moreover, 20 isolates out of 314 (6.4%; CI 95% = 3.7–9.1) showed an intermediate profile in response to the screening with CTX and CAZ and were also analysed genotypically (Figure 1).

All the *E. coli* ESβL isolated in pork meat samples were confirmed genotypically. The five ESβL + AmpC *E. coli* isolates were positive only for ESβL genes, apart from one isolate that also harboured AmpC genes from the *CMY* family and *FOX*. The phenotypically confirmed AmpC *E. coli* harboured ESβL genes, while none of them harboured AmpC genes. PCR analysis of phenotypically intermediate *E. coli* isolates showed that isolates harboured at least one ESβL-associated gene.

The ESβL genetically confirmed *E. coli* (*n* = 35) showed a high prevalence of the *bla*_CTX-M1_ gene (88.6%; CI 95% = 78.0–99.1), followed by *bla*_TEM_ (57.1%; CI 95% = 40.7–73.5), *bla*_SHV_ (34.3%; CI 95% = 18.6–50.0) and *bla*_CTX-M2_ (25.7%; CI 95% = 11.2–40.2).

Results from *E. coli* isolated from fresh wild boar meat showed that 11 out of 229 (4.8%; CI 95% = 2.0–7.6) were ESβL, 5 (2.2%; CI 95% = 0.3–4.1) were AmpC and 1 was both ESβL and AmpC (0.4%; CI 95% = 0–1.4). The intermediate profile identified during the screening with CTX and CAZ was detected in 26 isolates out of 229 (11.4%; CI 95% = 7.3–15.5) (Figure 2).

Nine out of 11 (81.8%; CI 95% = 59–100) ESβL and all the AmpC *E. coli* isolates were genotypically confirmed as ESβL. The isolate that was phenotypically confirmed as both ESβL and AmpC harboured only the *bla*_CTX-M1_ gene. None of the *E. coli* with an “intermediate” profile harboured the tested genes. The difference between the number of ESβL *E. coli* genotypically confirmed isolated from pork and from wild boar meat products was not statistically significant (*p* = 0.15).

The ESβL *E. coli* genetically confirmed (*n* = 15) showed a high prevalence of the gene *bla*_CTX-M1_ (93.3%; CI 95% = 80.6–100). On the other hand, for *bla*_TEM_ and *bla*_CTX-M2_, the prevalence was lower (6.7%; CI 95%= 0–19.6). None of the *E. coli* isolated from fresh wild boar meat harboured the *bla*_SHV_ gene.

The difference between the prevalence of genes *bla*_CTX-M1_ found in farm animals and that found in wild boars had a *p*-value of 0.12. On the other hand, prevalence of the *bla*_CTX-M2_ gene had a statistically significant *p*-value of 0.04 considering pigs vs. wild boars. A highly statistically significant *p*-value of 0.0004 was identified for the gene *bla*_TEM_. 

Genotyping patterns found in the isolates are reported in Table 1 and Table 2.

### 3.3. Colistin Resistance

A total of 314 *E. coli* isolated from pork raw meat samples and 229 isolated from fresh wild boar meat were tested to detect their susceptibility to colistin.

Phenotypically, none of the *E. coli* isolated from pork meat products were resistant to colistin. Looking at fresh wild boar meat, we found 4 *E. coli* isolates out of 229 (1.7%; IC = 95%, 0.05–3.4) to be resistant to the polymyxin drug. Two isolates showed an MIC value of 4 µg/mL, and the other two had an MIC value of 8 µg/mL. All the colistin-resistant isolates were genetically tested to evaluate the presence of the *mcr-1* gene, with negative results.

## 4. Discussion

Antibiotic resistance involves humans, animals and the environment and is currently one of the most important issues of the One Health paradigm [30]. Antibiotic-resistant bacteria from both animals and humans may be pathogenic to either species and transmitted by direct contact or through the food chain. 

In this study, AMR prevalence was confirmed phenotypically and genotypically. Phenotypically confirmed ESβL *E. coli* were present in 12/314 (3.8%) of sampled pork products. Prevalence reached 11.1% (35/314) following genotypic analysis. The first percentage is in agreement with data collected in the most recent EFSA report on antibiotic resistance in Europe [31]. The phenotypic data reported by 28 European Union member states and 3 other European countries (Norway, Iceland and Switzerland) in 2017 on ESβL-producing *E. coli* in pork meat are in fact quite encouraging. The prevalence was lower than in 2015 and varied from 11.1% in Malta to 0% in Luxembourg, Sweden, Finland, the United Kingdom, Iceland and Norway, with an average European value of 4.4% [31,32]. 

The Italian situation, in particular in Emilia Romagna and Lombardia (North Italy) showed a prevalence of ESβL *E. coli* isolated from pork fresh meat and meat products of 9.2% and 5.0%, respectively [33,34].

Moreover, the same EFSA report showed that 1.6% of *E. coli* isolated from pork meat in Europe in 2017 harboured a phenotypic AmpC resistance [31]. In our study, the prevalence (2.5%) was higher, but the genotypic analysis confirmed an encouraging 0.3% of *E. coli* isolated as AmpC. Only 0.3% of *E. coli* isolated from pork meat in Europe in 2017 showed both ESβL and AmpC resistance [31]. This percentage is lower if compared to the phenotypical analysis of the present study (1.6%), but it is in line with the results from the genotypic analysis. 

As previously mentioned, antibiotic-resistant bacteria can be widely disseminated in the environment via animal wastes [35], and wild animals might play a crucial role in the global spread of AMR. Animals are affected by the extensive use of antibiotics in human and veterinary medicine, although they are not directly treated with the molecules. Humans, in fact, influence habitats like cities, landfills and areas where intensive farming is practiced, producing important sources of resistant bacteria that can spread to wildlife and the environment [36,37,38]. For these reasons, wild animals are considered good indicators of environmental antibiotic resistance in specific areas [39]. 

ESβL-producing bacteria have been documented in various wild mammals and birds [36,40], and the ESβL genes found seem to reflect well the epidemiological situation tracked in humans and domestic animals [39].

Wild birds have the capacity to spread resistant bacteria over long distances. At the same time, wild boars, due to their close interaction with humans and livestock, their large home ranges and their eating behaviors, are considered important local sentinels for AMR surveillance in public health [39,41].

In fact, wild boars have acquired the behavior of approaching human and farm animal habitats in order to scavenge for food [42]. On the other hand, transmission of antimicrobial-resistant bacteria to food animals and humans is possible through contamination of food or water by wild animal manure, as reported by Greig et al. [43]. This could explain the contact between those two worlds and the possibility for resistant bacteria to influence the microbiota composition of different species, which then become carriers and dispensers of AMR [44].

All these hypotheses have led to a focus on wild boars as potential AMR promoters [44], suggesting that wild boar meat could represent a transmission source of resistant bacteria for consumers. In our study, when considering the genotypic analysis, the prevalence of ESβL *E. coli* was 6.5%, and two of the phenotypically confirmed *E. coli* did not harbour ESβL genes; at the same time, all the phenotypic AmpC *E. coli* were genetically confirmed as ESβL, and none had AmpC genes.

A recent study [45] evaluated the prevalence of cephalosporin-resistant *E. coli* isolated from the feces of deer, European bison and wild boars in Poland. The authors reported a prevalence of 1.7% of cephalosporin-resistant isolates from wild boars, with *bla*_CTX-M1_ and *bla*_CTX-M15_ as the most prevalent ESβL genes, while *bla*_CMY-2_ was the principal gene detected for AmpC. To the authors’ knowledge, the present study is the first to report the presence of ESβL/AmpC *E. coli* isolated directly from the fresh meat of wild boar. In fact, a recent paper published in 2017 [14] reported one *E. coli* harboring the *bla*_CTX-M1_ gene isolated from frozen red deer meat but none from wild boar meat. 

The most common genes encoding ESβL in animals are *bla*_CTX-M1_ and *bla*_CTX-M14_, followed by *bla*_TEM-52_ and *bla*_SHV-12_, while the gene mainly associated with resistance in AmpC-type β-lactamases is *bla*_CMY-2_ [7]. In enteric bacteria of domestic animals and large game animals in Europe, *bla*_CTX-M1_ is one of the most prevalent ESβL genes [36,46,47] and is also frequently found among *Enterobacteriaceae* recovered from the meat of farm animals [48,49]. In the present study, the *E. coli* isolated from both pork meat products and fresh wild boar meat frequently harboured *bla*_CTX-M1_, but only *bla*_CTX-M2_ and *bla*_TEM_ prevalence showed a statistically significant difference between pork and wild boar samples.

For decades, colistin has been the preferred treatment for intestinal infections in pigs, poultry, and cattle [12]. Its use has been strongly reduced in human medicine since the 1970s and in veterinary medicine since 2016 [50] due to increasing resistance to the molecule [51]. 

Data collected in this study showed that none of the *E. coli* isolated from pork meat products were resistant to colistin, highlighting that good farming practices are applied in the farms considered. Interestingly, looking at *E. coli* isolated from fresh wild boar meat, four isolates were found to be resistant to colistin; however, none of them harboured the *mcr-1* gene. This fact could be due to other less diffused colistin-resistance genes or to aspecific mechanisms [52].

Results from the profiles of *E. coli* judged “intermediate” for CTX and CAZ, according to EUCAST guidelines [26], were also interesting to consider. 

All the *E. coli* isolated from pork meat that were phenotypically intermediate were confirmed as ESβL when analysed genotypically (*bla*_CTX-M1_ and *bla*_TEM_ were the most prevalent genes), suggesting an underestimation from phenotypic data. 

Wild boar isolates, on the other hand, showed that none of the intermediate profiles were confirmed as ESβL genotypically. These results suggest that cephalosporin-intermediate expression mechanisms are aspecific in wild boar and not related to ESβL production. Moreover, it has not been scientifically proven that ESβL intermediate profiles can become resistant in the future (despite the absence of ESβL genes). Presumably, the lower selective pressure present in wildlife is the reason why “potential ESβL resistance” is not really expressed. 

The main limitations of this study reside in a noncomparable number of pork samples coming from different product types. In fact, each sample set (i.e., cotechino, sausages, meatballs, etc.) had a different number of samples, and for this reason it was not considered appropriate to compare resistance profiles among them. Additionally, the cutoff of colonies considered for analysis (10 CFU/g) may have led to underestimation of the “real” prevalence of resistant *E. coli.*

In conclusion, the study of the dynamics of AMR in domestic and wildlife environments showed that the wild boar is a good standard species as a sentinel for studying the phenomenon. *Sus scrofa* is an important reservoir for bacteria producing ESβL, and the control of these animals might represent a significant action for the surveillance of AMR transmission between urban and wildlife environments.

The selective pressure on microorganisms related to the resistance phenomenon found in fresh wild boar meat is lower than in pork products. The data obtained here show that phenotypic resistance in wild boar is not always reflected by the presence of genes, so it is also possible that the resistance is based on other aspecific mechanisms. 

Finally, when compared to previous studies, the resistance found in wild boar is not as alarming as that found in farming species, and the potential AMR health risk for consumers of pork and wild boar meat will necessarily need further investigations. 

## Figures and Tables

**Figure 1 microorganisms-09-00214-f001:**
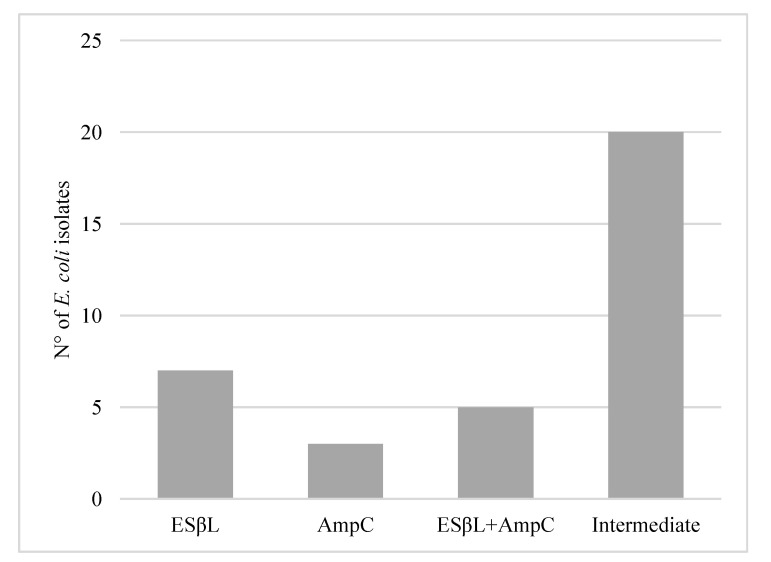
Number of ESβL, AmpC and ESβL + AmpC *E. coli* isolated from pork meat products and isolates with the intermediate profile in screening test for CTX and CAZ.

**Figure 2 microorganisms-09-00214-f002:**
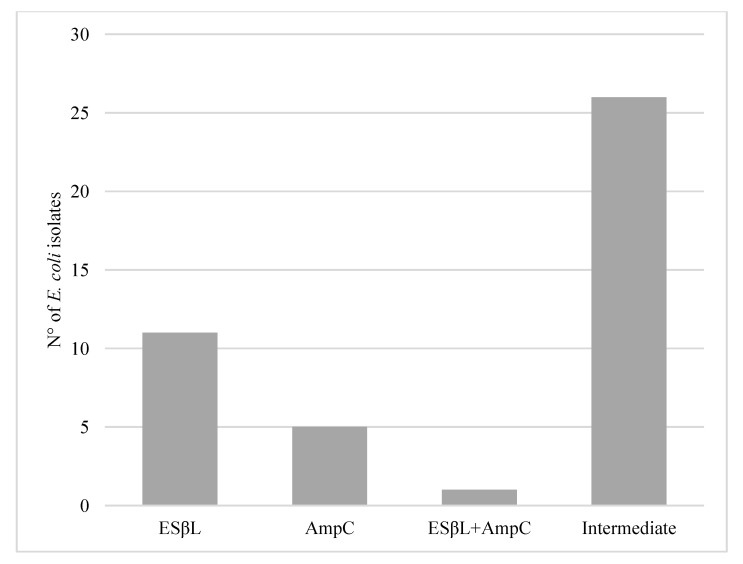
Number of ESβL, AmpC and ESβL + AmpC *E. coli* isolated from fresh wild boar meat samples and isolates with the intermediate profile in screening test for CTX and CAZ.

**Table 1 microorganisms-09-00214-t001:** Distribution and combination of ESβL and AmpC genes in *E. coli* isolated from pork and wild boar meat products (phenotypically confirmed ESβL, ESβL + AmpC and AmpC).

β-Lactamase Genes (ESβL and AmpC)	N° of *E. coli* Isolates from Pork Meat Product	N° of *E. coli* Isolates from Fresh Wild Boar Meat
***bla*** **_CTX-M1_**	2 (13.3%)	13 (76.4%)
***bla*** **_TEM_** ** + ** ***bla*** **_CTX-M1_**	1 (6.7%)	-
***bla*** **_SHV_** ** + ** ***bla*** **_CTX-M1_**	3 (20%)	-
***bla*** **_CTX-M1_** ** + ** ***bla*** **_CTX-M2_**	-	1 (5.9%)
***bla*** **_TEM_**	-	1 (5.9%)
***bla*** **_TEM_** ** + ** ***bla*** **_SHV_** ** + ** ***bla*** **_CTX-M1_**	1 (6.7%)	-
***bla*** **_SHV_** ** + ** ***bla*** **_CTX-M1_** ** + ** ***bla*** **_CTX-M2_**	3 (20%)	-
***bla*** **_TEM_** ** + ** ***bla*** **_SHV_** ** + ** ***bla*** **_CTX-M1_** ** + ** ***bla*** **_CTX-M2_**	4 (26.6%)	-
***bla*** **_TEM_** ** + ** ***bla*** **_SHV_** ** + ** ***bla*** **_CTX-M1_** ** + FOX + CMY (A** **mpC)**	1 (6.7%)	-
***No genes***	-	2 (11.8%)
***TOTAL***	15 (100%)	17 (100%)

**Table 2 microorganisms-09-00214-t002:** Distribution and combination of ESβL and AmpC genes in *E. coli* isolated from pork and fresh wild boar meat defined as “intermediate” in the screening test with CTX and CAZ.

β-Lactamase Genes (ESβL and AmpC)	N° of *E. coli* Isolates from Pork Meat Product	N° of *E. coli* Isolates from Fresh Wild Boar Meat
***bla*_CTX-M1_**	7 (35%)	-
***bla*_TEM_**	4 (20%)	-
***bla*_TEM_ + *bla*_CTX-M1_**	7 (35%)	-
***bla*_TEM_ + *bla*_CTX-M1_ + *bla*_CTX-M2_**	2 (10%)	-
***No genes***	-	26 (100%)
***TOTAL***	20 (100%)	26 (100%)

## Data Availability

Data sharing not applicable (all the data are shown in the paper).

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
