# Peer review of "Prevalence of ESβL, AmpC and Colistin-Resistant E. coli in Meat: A Comparison between Pork and Wild Boar"

_microorganisms, 2021, doi:10.3390/microorganisms9020214_

Round 1
Reviewer 1 Report
Rega et al aimed at defining the prevalence of ESβL, AmpC and colistin resistant E. coli in pork and wild boar meat products in order to define their potential transmission along the food chain.
Number of authors is appropriate for the information presented and references are up to date. Methods are well described and results are interesting.
A few comment, questions and suggestions:
Did authors find any difference in resistance based on the different type of products (sausages, cotechino, salami, meatballs or meat skewers)?
The statistical analysis paragraph as a stand alone is confusing and I would suggest to include those along the results as you reported with the p value.
Please add limitations and strengths of your study in the manuscript.
If these testing becomes part of the standard, would these products that are contaminated held and discarded so they do not get to the public. Please expand on the potential actionable items of these findings in avoiding transmission of resistance through the food chain.
Please address/correct the following:
Introduction-
"It is estimated that resistance to second and third generation of broad-spectrum cephalosporins will double by 2030" (all third generation cephalosporins are broad spectrum).
"E. coli become resistant to cephalosporins thanks due to their ability to produce extended spectrum β-lactamases (ESβL)" (ESBL stands for Extended Spectrum Beta lactamases, not just B lactamases).
"The mechanism of resistance is based on the transfer of PEtN (a transferase enzyme) to lipid A..." change to
"The mechanism of resistance is based on the transfer of phosphoethanolamine (PEtN) transferase to lipid A..."
Please review the following sentence:
"The mechanism seems to be related to the expression of mcr genes [10], generally located on plasmids, and many gene variants have been identified (mcr-1 to mcr-9) [12] "
p values should not be capitalized.
Author Response
Reviewer 1:
Rega et al aimed at defining the prevalence of ESβL, AmpC and colistin resistant E. coli in pork and wild boar meat products in order to define their potential transmission along the food chain.
Number of authors is appropriate for the information presented and references are up to date. Methods are well described and results are interesting.
A few comment, questions and suggestions:
Did authors find any difference in resistance based on the different type of products (sausages, cotechino, salami, meatballs or meat skewers)? The number of product types was very different and the authors felt any type of comparative analysis would not be reliable. These considerations have been added to the limitations of the present study (please see page 9).
The statistical analysis paragraph as a stand alone is confusing and I would suggest to include those along the results as you reported with the p value. We modified it, thank you for the suggestion.
Please add limitations and strengths of your study in the manuscript.
The following has been added to the revised manuscript: “The main limitations of this study mainly reside in a non-comparable number of pork samples coming from different product types. In fact, the number of each sample set (i.e. cotechino, sausages, meatballs, etc) was not similar and for this reason it was not con-sidered appropriate to compare resistance profiles among them. On the other side, also the cut-off of colonies considered for analysis (10 CFU/g) may have led to underestimate the “real” prevalence of resistant E. coli.”
The major strength of the present study is the first report of the presence of ESβL/AmpC E. coli isolated directly from the fresh meat of wild boar. This is already included the manuscript (see page 8).
If these testing becomes part of the standard, would these products that are contaminated held and discarded so they do not get to the public. Please expand on the potential actionable items of these findings in avoiding transmission of resistance through the food chain.
The authors agree that the transmission of resistance through the food chain must be avoided. The present study was aimed at highlighting the prevalence of a phenomenon, that must necessarily be controlled primarily at the level of primary production, through awareness of the proper use of the antimicrobials. The authors feel that eventual standardization of resistance analyses before marketing meat products would be very complex and too ambitious action to propose.
Please address/correct the following:
Introduction-
"It is estimated that resistance to second and third generation of broad-spectrum cephalosporins will double by 2030" (all third generation cephalosporins are broad spectrum). Done
"E. coli become resistant to cephalosporins thanks due to their ability to produce extended spectrum β-lactamases (ESβL)" (ESBL stands for Extended Spectrum Beta lactamases, not just B lactamases). Done
"The mechanism of resistance is based on the transfer of PEtN (a transferase enzyme) to lipid A..." change to
"The mechanism of resistance is based on the transfer of phosphoethanolamine (PEtN) transferase to lipid A..." Done
Please review the following sentence:
"The mechanism seems to be related to the expression of mcr genes [10], generally located on plasmids, and many gene variants have been identified (mcr-1 to mcr-9) [12] " Done
p values should not be capitalized. Done

Reviewer 2 Report
The misuse and overuse of antibiotics in livestock have led to the spread of resistant bacteria within the meat product. In this manuscript, Rega and coworkers evaluate the prevalence of ESBL/AmpC/mcr producing E.coli isolated from pork and wild boar meat products. This work will be of broad interest to readers of Microorganisms journal. However, some items need to be addressed before publication.
- Page 1, E.coli needs to be italicized in the abstract section.
- Page 1, last sentence: EsβL stands for extended extended-spectrum beta-lactamase. The authors need to rephrase the sentence.
- Page 3: It is essential to know the source of meat samples in the “sample collection” section. Are they from meat processing companies, slaughterhouses, grocery stores, or farms?
- Page 4, last paragraph: authors claim that one isolate harbored AmpC genes CTM and FOX. Hanson et al. described six sets of Ampc-specific primers in reference 28, which do not contain CTM. The author should clarify it.
- Page 5: It would be nice if authors can test the existence of mcr variants since mcr-1 result is negative.
Author Response
The misuse and overuse of antibiotics in livestock have led to the spread of resistant bacteria within the meat product. In this manuscript, Rega and coworkers evaluate the prevalence of ESBL/AmpC/mcr producing E.coli isolated from pork and wild boar meat products. This work will be of broad interest to readers of Microorganisms journal. However, some items need to be addressed before publication.
- Page 1, E.coli needs to be italicized in the abstract section. Done
- Page 1, last sentence: EsβL stands for extended extended-spectrum beta-lactamase. The authors need to rephrase the sentence. Done
- Page 3: It is essential to know the source of meat samples in the “sample collection” section. Are they from meat processing companies, slaughterhouses, grocery stores, or farms? Done
- Page 4, last paragraph: authors claim that one isolate harbored AmpC genes CTM and FOX. Hanson et al. described six sets of Ampc-specific primers in reference 28, which do not contain CTM. The author should clarify it. Answer: The authors mistakenly wrote the name of the primer used instead of the genes. The proper corrections have been done.
- Page 5: It would be nice if authors can test the existence of mcr variants since mcr-1 result is negative. Answer:The authors agree. Indeed, the study focused on the most prevalent gene involved in colistin resistance identified in Europe, for this reason we chose to look only for mcr-1. Unfortunately, the needed primers are not currently available to us, but If the reviewer deems it necessary to perform these analyses, we would need more time for revision (1 month) than currently granted, in order to purchase primers and process the four phenotypically colistin resistant strains. The Authors look forward to the Editor’s response.

Round 2
Reviewer 2 Report
The authors have addressed my concerns. The revised manuscript has been significantly improved.